# Intra-Plot Variable N Fertilization in Winter Wheat through Machine Learning and Farmer Knowledge

Asier Uribeetxebarria, Ander Castellón, Ibai Elorza and Ana Aizpurua *

NEIKER-Basque Institute for Agricultural Research and Development, Berreaga 1, 48160 Derio, Biscay, Spain
* Correspondence: aaizpurua@neiker.eus; Tel.: +34-666-451-862

**Abstract:** The variable fertilization rate (VFR) technique has demonstrated its ability to reduce nutrient losses by adapting the fertilizer dose to crop needs. However, transferring this technology to farms is not easy. This study aimed to make a variable fertilization map in a commercial plot where there is no data from a yield monitor, combining machine learning techniques and farmer's knowledge. In addition to the normalized difference vegetation index (NDVI) obtained from Sentinel-2 and a digital elevation model (DEM), information captured by a yield monitor in 2019 was used to train and validate models. Among the 15 algorithms trained, the best result was obtained by the random forest (RF), with an RMSE of 496 and $R^2$ of 0.90. Using the "leave one out" technique, the capacity to predict an entire plot was tested. Finally, the RF algorithm was tested on a 12-hectare wheat plot where no yield data were available. The novelty of this work lies in the collaborative work developed between farmers and researchers to implement the VRF technique in plots where precise yield data do not exist and in the "leave one out" validation. The collaboration between scientists and farmers resulted in a very positive exchange of information that allowed the farmer to change the fertilization strategy of the whole farm and the scientists to better understand how soil properties and plot history affect yield.

**Keywords:** precision agriculture; nitrogen fertilization; NDVI; on-farm experimentation; yield prediction





## 1. Introduction

Farmers and agronomic advisors, together with scientists, must be at the center of the innovation process if the social, ecological, and economic objectives set by the EU are to be achieved. Collaborative work between these groups would allow the development of innovative solutions. Farmers are custodians of local knowledge and practices [1], and experimentation is part of their daily routine due to their need to improve the quality and productivity of their crops [2], even if, normally, a proper measurement is not made. On the other hand, scientists, when conducting agronomic trials, spend many hours measuring and controlling variables of interest in small experimental fields, and for this purpose, they use a wide variety of sensors. In this way, the effects of the tested factors can be correctly measured and thus demonstrate their relevance. Additionally, these trials are often not representative of commercial plots, as they are not able to consider spatial variability adequately. In many cases, the differences generated by plot variability outweigh the effect of the treatment itself [3]. Thus, the methodology being used by scientists has been one of the reasons for the lack of interaction between the two groups. Collaborative work between scientists, capturing and interpreting the data, and farmers with a deep knowledge of their plots is therefore essential. Unfortunately, there is a lack of flow of the knowledge generated at both levels, the professional and the scientific, and the interaction between the two is not always productive [4]. Because of this, the on-farm experimentation (OFE) strategy is presented as an alternative to solve the existing problems between the two groups through advancement to more modern and collaborative farming [5]. In summary, OFE is a global

trend that recognizes the collaborative work between farmers and scientists as a pathway to develop new innovative ways to solve the challenges facing modern agriculture [6].

Precision agriculture is a management strategy that considers temporal and spatial variability to improve the sustainability of agricultural production, and its philosophy is in line with that of OFE. Adequate fertilization of crops is one of the greatest challenges faced in modern agriculture. More specifically, a good fertilizer strategy will have to consider potential yield variability and adjust the applied dose accordingly [7], applying more fertilizer in the more productive areas and less in the less productive ones. Since crop fertilizer demand is related to crop yield [8], it is essential to adjust the N fertilizer supply to crop needs. A lower than required dose reduces production, as does an excessively high dose, since it produces physiological damage [9]. Additionally, nitrogen fertilizers applied in excessive quantities cause N losses, mainly through ammonia volatilization, denitrification, and leaching [10]. However, the most common practice consists of homogeneous fertilization for the whole plot, without considering the different requirements of the crop that depend on the intrinsic variability of the plot. Accurate and early yield estimation is a good way to adapt fertilization to plant requirements. This is because the yield and therefore the amount of N needed varies from year to year. Thus, if it is possible to estimate the yield early, before applying the last N dose, the fertilizer contribution can be decided for each campaign based on the needs of each plot or even the area of the plot. Therefore, in order to be useful, yield estimation must be done before the application of the cover crop fertilizer. In the study area, this is carried out at the beginning of stem elongation (GS 30) [11] for cereal cultivation.

Thus, through variable-rate fertilization (VRF), precision farming enables the adjustment of fertilizer application to crop needs [12]. The first step to applying variable rate fertilization consists of delineating yield zones. For this purpose, one or more variables related to yield (vegetation index, elevation, soil properties, etc.) are selected [13] and a model is then established that relates the variables with yield, which is the property to be estimated. Subsequently, based on the N extractions per unit produced, the fertilization maps are calculated, using an unsupervised multivariate classification algorithm, and auxiliary variables are classified, creating the site-specific management zones [14]. Many farmers of the Basque Country (Spain) have the machinery and devices required to apply the VRF technique, but the lack of technical knowledge to make fertilization prescription maps is an impediment to its implementation.

Vegetation indices can be obtained from multispectral satellite images. The recent launches of the Sentinel-2 (S2) satellite constellation by the European Space Agency (ESA) may potentially increase the adoption of precision agriculture techniques by small- and medium-sized farmers [14]. Two twin S2 satellites (A + B) have been specifically designed to meet the needs of the agricultural community and researchers. The images, offered free of charge by the ESA through its Copernicus platform, have a high spatial resolution, a high number of multispectral bands, and a short revisit period [15]. To monitor crop cover, yield, and health, visible (blue, green, and red) bands and near-infrared (NIR) regions of the electromagnetic spectrum are used [16]. Strong absorption of red energy and strong reflection of NIR energy are characteristics of healthy crops [17].

In recent years, the application of machine learning algorithms in precision agriculture has increased sharply [18]. It is interesting to establish models that estimate yield from easily obtainable auxiliary variables such as vegetation indices, terrain elevation, etc. Sensor data analysis using machine learning algorithms has made it possible to handle complex interactions that happen in commercial plots, enabling data-driven decision-making. A combination of different data sources and a machine learning model to obtain high-precision yield maps was used by Kamir et al. [19]. Hunt et al. [20] utilized the data obtained from an optical satellite and environmental information to forecast wheat yield. If the use of yield monitors was more widespread, it would be possible to train and test algorithms to make an early yield estimation. Unfortunately, the use of yield monitors

is not widespread in Europe [21], which hinders the collection of data to train and test algorithms on commercial farming plots.

The objective of the present study was to make variable nitrogen fertilization maps in plots without previous yield information. For this purpose, a methodology was developed that allows combining the farmer's knowledge with the use of available information—in this case yield, the normalized difference vegetation index (NDVI), and a digital elevation model (DEM)—and its statistical analysis.

## 2. Materials and Methods

### 2.1. Field Sites

This study was conducted using data from 14 rainfed wheat fields of Araba/Álava (northern Spain), belonging to two farmers. Farmer 1 had 13 plots and a yield monitor, while farmer 2 did not have a yield monitor and wanted an N fertilization map for use at one of his plots. The average size of the plots analyzed was 4.3 ha. The 12.2 ha Torres plot is the biggest while the smallest is Apelarri F with 2.1 ha (Table 1). The extension covered in this study was 60.4 ha (Figure 1). All plots were sown at a seed rate of 230 kg ha$^{-1}$ between 19 and 25 November 2018 (Table 1). The same fertilization scheme was applied for all plots of Farmer 1. The basal application of fertilizer was 53 kg N ha$^{-1}$, 36 kg P ha$^{-1}$, and 102 kg K ha$^{-1}$ and was applied on 30 December 2018. The second and third N topdressing fertilizer applications were made on 26 February 2019 and 25 March 2019. Calcium ammonium nitrate (CAN), which has a nitrogen concentration of 27%, was used for this purpose. The CAN fertilizer dose applied was 220 and 210 kg ha$^{-1}$. The total N rate was 169 kg N ha$^{-1}$. The yield information obtained from a yield monitor from the 13 plots of the 2018–2019 cropping season was used. In the 2019–2020 cropping season, the plot of farmer 2 was sown at a seed rate of 230 kg ha$^{-1}$ and basal fertilization consisted of 412 kg ha$^{-1}$ of 15-13-13 (N-P$_2$O$_5$-K$_2$O). Normally, a herbicide treatment is carried out approximately in GS31 and a fungicide treatment about 15 days later.

**Table 1.** Average plot yield, area, number of Sentinel-2 pixels, and elevation.

| Plot | Yield (t ha$^{-1}$) | Area (ha) | S2 Pixels within Plot | Elevation (m) |
|---|---|---|---|---|
| Alto | 8.6 | 5.1 | 426 | 502 |
| Apelarri | 8.2 | 2.6 | 207 | 505 |
| Apelarri F | 7.8 | 2.1 | 185 | 508 |
| Babea | 6.7 | 3.8 | 323 | 521 |
| Baratua | 5.6 | 2.7 | 217 | 511 |
| Foronda | 6.4 | 3.2 | 254 | 513 |
| Iruleku | 7.4 | 4.1 | 346 | 534 |
| Kukura | 6.3 | 5.0 | 417 | 508 |
| Menor | 5.2 | 4.6 | 358 | 538 |
| Ollavarre | 4.6 | 4.3 | 353 | 554 |
| Otatza | 6.3 | 3.0 | 246 | 541 |
| Parque | 4.7 | 5.2 | 413 | 531 |
| Prado | 7.1 | 2.5 | 208 | 501 |
| Torres | | 12.2 | 916 | 511 |

The average elevation of the plots ranged between 480 and 560 m above sea level. Annual rainfall is usually around 750 mm. Summers are mild, with an average temperature in the months of July and August of 20 °C due to the influence of cold ocean currents, while December, January, and February are milder (6 °C) than in other climates in similar latitudes. According to Köppen [22], the region is classified as having a "warm-summer Mediterranean climate" (Csb). The soils in this study have carbonates with an alkaline pH, normally between 8 and 8.5. Regarding texture, the predominant fraction is generally silt. In the case of the implementation plot, the texture was silty loam. The rotation normally includes wheat, 1 or 2 years of barley, and a non-cereal crop. This crop can be an oilseed, protein, or forage crop. In the irrigated plots, the rotation is wheat, barley (1 year), and

an extensive irrigated crop: potato, sugar beet, green beans, or corn. In the best soils, the alternative is wheat-irrigated crop.

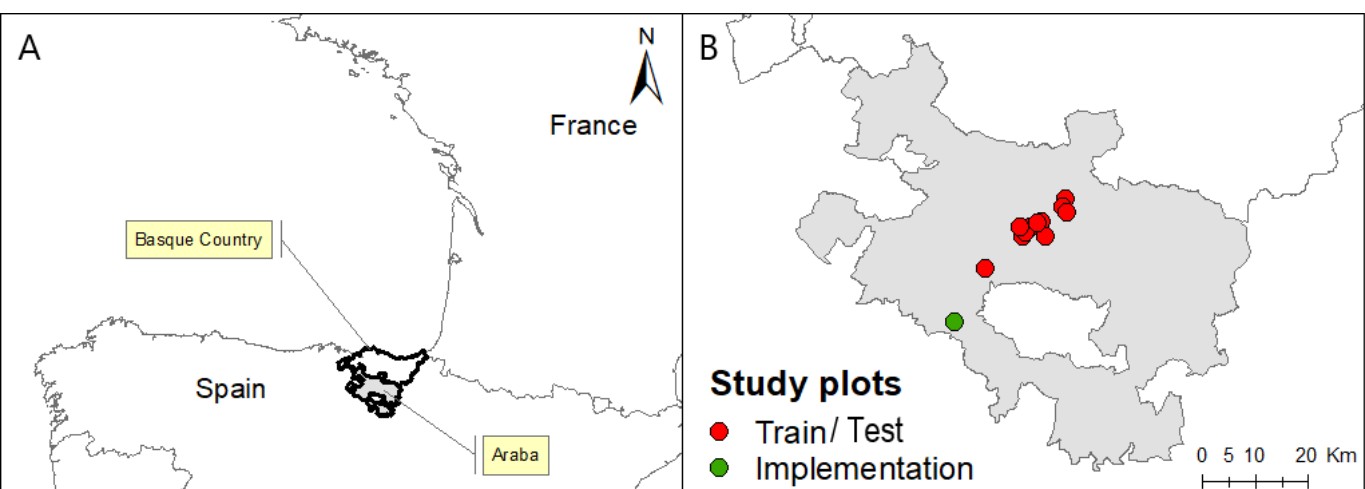

**Figure 1.** (**A**) General location of the province of Araba/Álava in Spain. (**B**) Location of 14 wheat plots. The red dots represent the plots belonging to farmer 1 used to train and test the algorithm. The green dot represents the implementation plot (farmer 2).

*2.2. Working Procedure*

The first step was the estimation of the yield to adjust the fertilizer dose to the crop requirements. For this purpose, 15 machine learning algorithms were used (Figure 2). These algorithms were trained using data from a yield monitor installed on a harvester (variable to be predicted), 14 predictor variables obtained from the S2 satellite (NDVIs from February to June), and one from a DEM. Data were collected in 13 agricultural plots (Figure 1B, red dots). Seventy percent of all the data collected were used to find the algorithm with the best wheat yield prediction ability. The training dataset was randomly selected and comprised information of all plots. The five best-performing algorithms were then validated using the remaining data (30% of the dataset), with the aim of finding the one that had the best predictive capability.

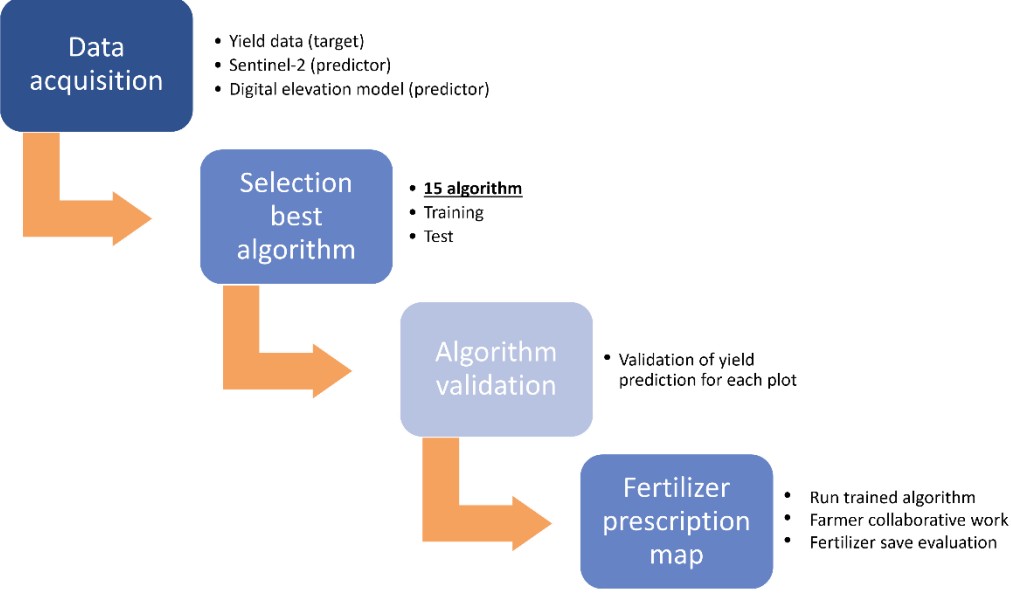

**Figure 2.** Flowchart of the proposed working methodology to obtain a fertilizer prescription map in one agricultural plot without yield data.

In the next step (Figure 2), the predictive capacity of the selected algorithm was evaluated for a whole plot. For this purpose, the algorithm was retrained using all the data except the data of the plot to be predicted. This process was repeated for each of the 13 plots (Figure 1B, red dots).

Finally, the algorithm was run using auxiliary data (S2, DEM) from farmer 2, who did not have a yield monitor but wanted to adjust the fertilizer rate. The result of the prediction map was compared with the farmer and, using this map as a baseline, a fertilizer prescription map was made together with him.

### 2.3. Yield Data

The high-resolution yield data used to train and test the regression machine learning algorithm belong to the 13 wheat plots of farmer 1 (Figure 1B, red dots). Data were collected by combining a yield monitor with a GPS mounted on a John Deere T560 combine harvester. The working width of the harvester was 6.1 m. The GPS receiver works with RX corrections, allowing it to work with a 15 cm precision. Wheat yield data were acquired during the harvest period of 2019, between 23 July and 9 August. Before being usable, the yield data had to be pre-processed to remove inaccurate grain yield measurements. The cleaning process consists of applying different rules to eliminate outlier measurements. The measurements located at a distance shorter than 15 m from the plot edge were eliminated to avoid the border effect. Finally, using the ordinary kriging technique, the data were interpolated. A more detailed explanation about the process followed to pre-process the dataset can be found in Uribeetxebarria et al. [23]. The average yield of the 13 plots obtained after performing this process ranged between 4.6 and 8.6 t ha$^{-1}$ (Table 1).

### 2.4. Normalized Difference Vegetation Index (NDVI) and Digital Elevation Model (DEM)

The NDVI was selected as the reference index due to its prior use in many wheat studies [24], and because its relationship with wheat yield was previously confirmed by Aranguren et al. [25] for the study region. The NDVI was calculated as (Equation (1)):

$$NDVI = (NIR-RED)/(RED + NIR) \tag{1}$$

where NIR is the near infrared wavelength and RED is the red wavelength. The spatial resolution of both bands is 10 m. While the RED band is centered at 665 nm and its width is 30 nm, the NIR band is centered at 842 nm and its width is 105 nm.

All cloud-free S2 images with 2A processing level (bottom of the atmosphere) [26] from 8 February and 28 June 2019 were downloaded from the Copernicus Open Access Hub platform (https://scihub.copernicus.eu/ (accessed on 22 September 2022)). Thus, 14 sequential images were used in the analysis, encompassing the period of the crop cycle that has active vegetation and therefore covering the most relevant wheat development phases (GS21, GS30, GS39, and GS85, according to the Zadoks scale). The images between 8 February and 20 March correspond to the tillering growth phase, those between 30 March and 14 May to the stem elongation growth phase, and those between 15 May and 18 June coincide with the heading growth phase. Finally, the images from June 9 to June 28 represent the ripening growth phase. Each experimental plot used in this study was within the S2 30TWN tile. As the images used were at processing level 2A, the step to correct 1C images (atmospheric effect correction) was not necessary. A 15 m buffer was applied to each plot to ensure that all S2 pixels were completely within the plot. Table 1 shows the number of pixels within each parcel.

A DEM, derived from a LiDAR flight made in 2016, was used to extract the elevation of each pixel (Table 1). This information is available at Geoeuskadi, the official spatial information repository of the Basque Country (https://www.geo.euskadi.eus/s69-bisorea/es/x72aGeoeuskadiWAR/index.jsp (accessed on 22 September 2022)). The final dataset was composed of 4869 instances.

*2.5. Machine Learning Regression Algorithm*

It is common in machine learning to test different algorithms to find the one that best meets the needs of the user. In this specific study, the variable to be predicted was wheat yield at the end of the season. The auxiliary variables used were 14 NDVI images obtained from each of the S2 images, and the elevation obtained from the DEM. All variables were introduced into the model at the same time. With this inquiry in mind, the predictive ability of 15 different regression algorithms was analyzed using the Caret package [27]. The algorithms used were as follows: (1) Simple linear model (LM MDT), with DEM as the independent variable, (2) Artificial Neural Networks (NNET), (3) Partial Least Squares (PLS), (4) Linear Regression with Forward Selection (LeapForward), (5) Linear Model (LM), (6) General Linear Model (GLM), (7) Lasso (Lasso), (8) Bagged CART (Treebag), (9) Bagged EARTH (BagEarth), (10) Bayesian Regularized Neural Networks (Brnn), (11) Gaussian Process (Gaussian), (12) Boosted Tree (BstTree), (13) Support Vector Machines with Radial Basis Function Kernel (SvmRadial), (14) k-Nearest Neighbors (KNN), and (15) Random Forest (RF). Once the pre-processed dataset was standardized, it was used to train all the algorithms. The cross-validation (*n* = 10) resampling technique was used to confirm the stability of each algorithm. The hyperparameters of the different algorithms were adjusted using the Caret package implemented in R.

Seventy percent of the data (2641) were used to train all the algorithms. The accuracy of each model was evaluated using the root mean square error (RMSE) and the coefficient of determination (R2). The five models that obtained the best results in the training phase were selected for evaluation of their predictive capability with the test dataset (the remaining 30% of the dataset). This dataset includes the same variables as the training dataset, but the yield variable was used to validate the model.

Once the algorithm with the best result was chosen, it was validated using the "leave one out" technique. In this case, the technique consists of training the model using all the data from all the plots except one. Then, the algorithm was validated using information from the remaining plot. This process was repeated for all the plots (13 times).

*2.6. Statistics*

The performance of the models was compared using the RMSE and the $R^2$. The ordinary kriging method was used to interpolate yield data and ISODATA, an unsupervised classification algorithm, based on the *k-means* algorithm was used to cluster the images.

### 2.6.1. Root Mean Square Error (RMSE)

This statistic is often used to measure the differences between values (sample or population values) predicted by a model and an estimator of the values observed (Equation (2)):

$$RMSE = \sqrt{\sum_{i=1}^{N} \frac{(Ei - Oi)^2}{n}} \tag{2}$$

where *O* represents the observed values, *E* is the estimated values, and *n* is the number of samples.

The RMSE is a measure of accuracy to compare the forecasting errors of different models for a particular dataset and not between datasets, as it is scale-dependent. The lower the RMSE, the better the model fits the dataset.

### 2.6.2. Coefficient of Determination ($R^2$)

This statistic is used in the context of statistical models whose main purpose is the prediction of future outcomes based on other related information. It supplies a measure of how well-observed outcomes are replicated by the model, based on the proportion of total variation of results explained by the model [28].

### 2.6.3. Interpolation

The use of geo-referenced information in models (satellite pixels and yield monitors) makes it possible to construct potential yield maps. However, the output of this type of model is a vector with no spatial dimensions, which produces a map of points, leaving areas of the map without information. To facilitate the interpretation of the information by the end user, a continuous map without gaps should be created, and the most common technique for that is an interpolation. The "ordinary kriging" method was used to interpolate the data. For that, a semi-variogram was fitted using the rational quadratic function. The interpolation showed a strong autocorrelation (<0.25) [29], which indicates a good quality of the map. The higher the spatial autocorrelation, the better the spatial estimation [30].

### 2.6.4. ISODATA

The procedure for delineating site-specific management zones relies on an iterative algorithm that begins by assigning an arbitrary mean to each class. Then, the pixels are reallocated to each group based on the minimum Euclidean distance between each pixel value and the mean value of each group. The groups are defined when the maximum number of iterations is reached or when the number of pixels changing from one category to another does not exceed a predefined threshold [31].

## 3. Results and Discussion

### 3.1. NDVI Evolution and Correlation with Yield

Figure 3 shows the changes over time of the NDVI in the 14 study plots, which was similar to that reported by Magney et al. [32]. The onset of each phenological period was determined using the STICS (Simulateur multidisciplinaire pour les Cultures Standard) software [33], as Zadoks growth stages were not determined in the field. The stem elongation phase (GS31), which lasted until 30 March, was characterized by a rapid increase in NDVI (Figure 3). In this phase, three plots (Parque, Menor, and Baratua) showed a lower NDVI value and, at the end of the crop season, their yields were among the lowest (Table 1). The heading phase (GS39) was characterized by a slow NDVI increase and lasted until 14 May, when the highest values were reached. The maximum NDVI values ranged from 0.91 for the Alto plot to 0.70 for the Menor plot. The Alto plot was the most productive (8.6 t ha$^{-1}$) and the Menor plot the least (5.3 t ha$^{-1}$) (Table 1). From 13 June onwards, the NDVI drops rapidly, showing that the crop was in the ripening stage (GS85) in each plot.

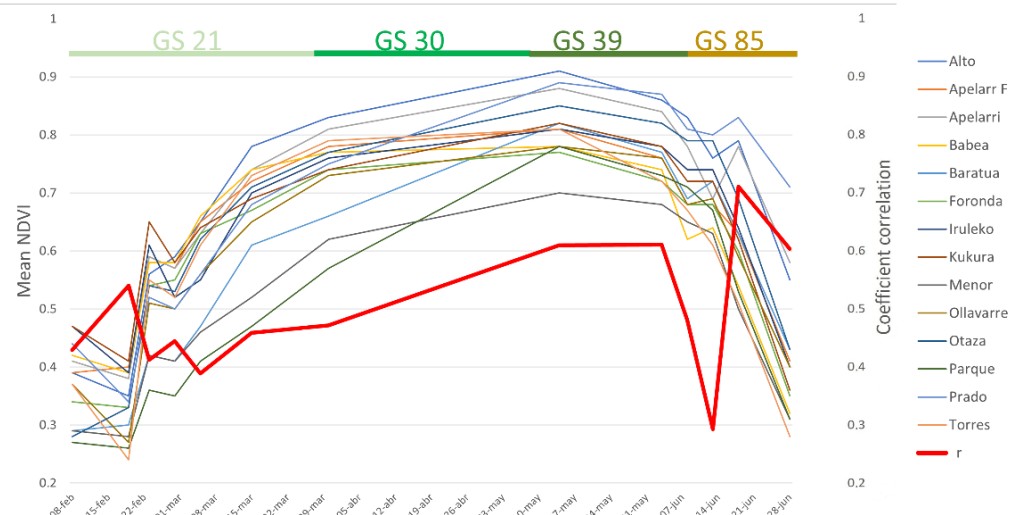

**Figure 3.** Seasonal trend in NDVI data of the 14 plots over the 2019 season. The thick red line shows changes in the Pearson correlation coefficient (r) between NDVI and yield (Torres plot not included). GS refers to the wheat growth stage. GS21 Tillering, GS31 Stem elongation, GS39 Heading, and GS85 Ripening.

Figure 3 shows the NDVI evolution for each plot. In addition, the Pearson correlation (thick red line) between the mean NDVI of all plots and the final yield is also represented. This correlation is similar to the one reported by Martí et al. [34], with a modest increment during the tillering phase (GS21) and a steady increase from the stem elongation stage (GS31) until the end of heading (GS59). Surprisingly, before reaching the maximum correlation coefficient on 18 June, the correlation collapsed on 13 June (Figure 3). The NDVI value for this date also underwent a slight drop (Figure 3). The loss of correlation may be due to a failure in the NDVI reading. This relationship between NDVI and yield supported the use of NDVI as one of the auxiliary variables for yield estimation. The climatic variables affecting crop development were not considered in this study as the plots were close to each other and the data from only one campaign were analyzed.

The work in collaboration with the farmer started at this point. The NDVI images corresponding to the highest correlation with yield were analyzed together with the farmer to better understand the underlying reasons for yield variability. The farmer confirmed that, in general, the NDVI was a good representation of what had happened that season with the crop but also pointed out that, in some cases, it was not representative of most years. For example, the river next to one parcel had overflowed, flooding the crop. The NDVI showed that production in that area was low, but the farmer knew that it was usually the most productive area as it had deep and fertile soils. Therefore, the use of vegetation indices such as the NDVI can be of value for the design of fertilizer management zones, but, if fertilization is to be adjusted using a yield map from the previous season, it is essential to ensure that it is representative of the average yield of the plot and its variability.

### 3.2. Selecting the Best Machine Learning Regression Algorithm for Wheat Yield at S2 Pixel Resolution

The comparative analysis of the performance of the 15 algorithms with the training dataset shows that RF had the best results (Figure 3). The RMSE values ranged from 1325 for the linear model to 528 for the RF. The $R^2$ ranged from 0.39 for the linear model with DEM information to 0.90 for the RF. The standard deviation of LM with DEM (LM MDT) is greater than the LM algorithm (Figure 4). These results suggest that the data used to train the model (in each iteration) had a greater influence on the results. Considering this, it is not strange that, although the RMSE of the LM MDT is lower, the $R^2$ is somewhat lower in relation to the LM. One of the reasons for the better performance of RF is that it is optimized to work with many variables and is difficult to overtrain [35]. Furthermore, RF is not strongly affected by the presence of outliers, as these are clustered and subsequently analyzed. Each tree in the RF has high variance and low bias, but it is reduced by averaging all tree results. In this case, the algorithm was composed of 500 trees. Finally, RF does not require much model training time because it works with subsets rather than whole trees.

Surprisingly, the NNET algorithm, widely used in precision agriculture to estimate the yields of different crops [36,37], performed poorly in this experiment (Figure 4). This could be because with medium-sized datasets, such as the ones used in this article, the NNET does not train itself adequately because it needs huge amounts of data to work properly [38].

The KNN algorithm is another widely used algorithm for yield estimation. However, in this experiment, it was not among the top five algorithms. Its low performance was probably a consequence of the use of many correlated variables [39].

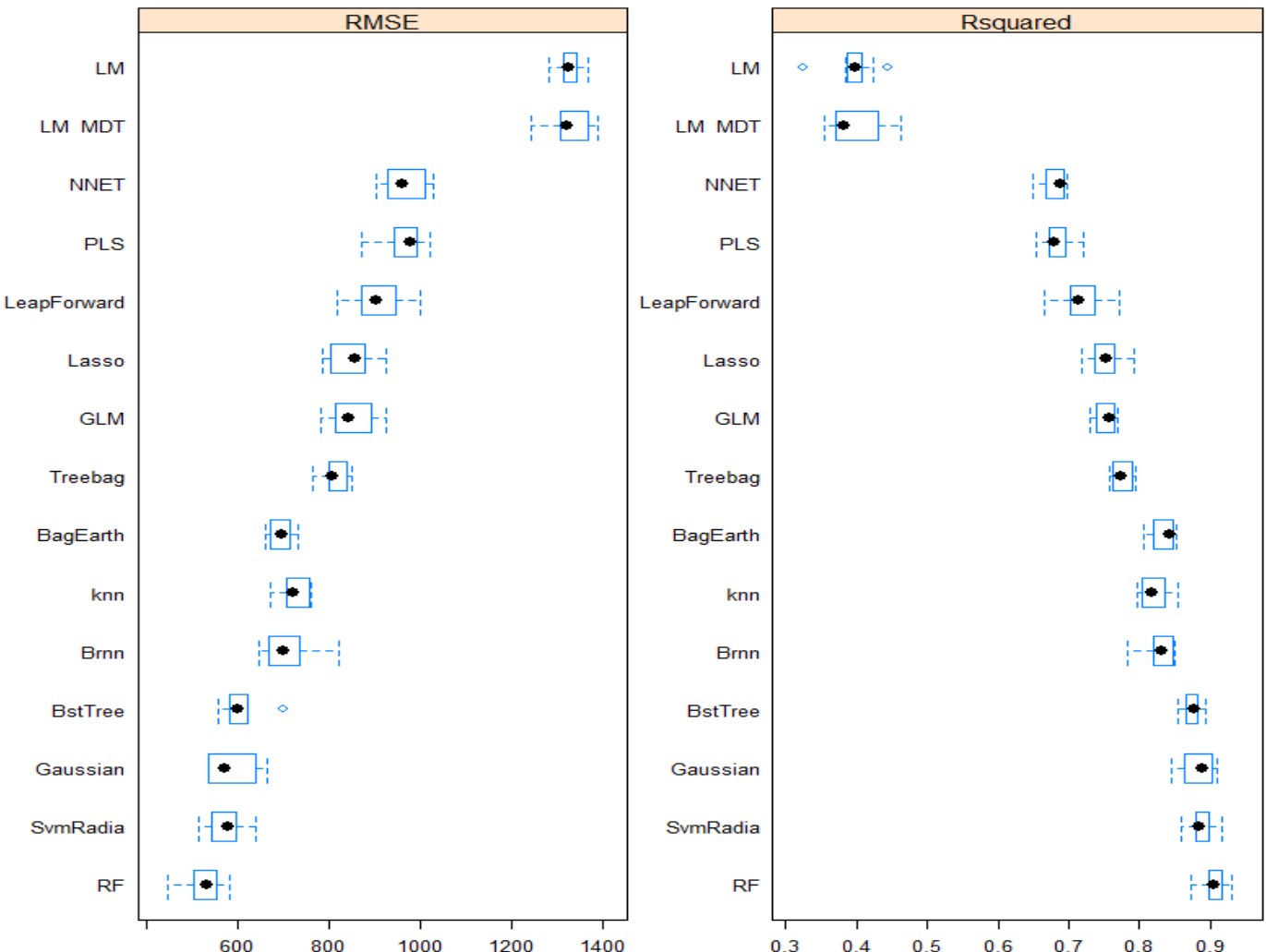

**Figure 4.** Comparison of the performance of the 15 machine learning algorithms, ordered from lowest to highest $R^2$ and RMSE (LM = Linear model, LM MDT = Simple linear model with DEM, NNET = Artificial Neural Networks, PLS = Partial Least Squares, LeapForward = Linear Regression with Forward Selection, Lasso = Lasso, GLM = General Lineal Model, Treebag = Bagged CART, BagEarth = Bagged EARTH, knn = k-Nearest Neighbors, Brnn = Bayesian Regularized Neural Networks, BstTree = Boosted Tree, Gaussian = Gaussian Process, SvmRadia = Support Vector Machines with Radial Basis Function Kernel, RF = Random Forest). The mean value of 10 iterations is represented with a black dot, the blue box represents the standard deviation, and the blue lines represent the threshold of the outlier.

The five algorithms (RF, SvmRadial, Gaussian, BstTree, and Brnn) that obtained the best results with the training dataset were validated using the test dataset. The prediction capability of the five selected algorithms was significant ($p < 0.001$). The best results were obtained again by the RF algorithm, with an $R^2$ value of 0.90 and an RMSE of 496. On the other hand, Brnn was the algorithm with the worst results obtained with the test dataset ($R^2 = 0.81$ and RMSE = 722). The rest of the algorithms performed practically equally. The RMSE ranged between 589 and 573, and the $R^2$ was the same: 0.86. The RF algorithm was used by Hunt et al. [30] with good results to predict wheat yield for large areas of the United Kingdom. Taking both statistics (RMSE and $R^2$) into account, and the fact that it has been used successfully for similar purposes before, the RF was selected to analyze its performance when predicting the wheat grain yield of the whole plot of farmer 2, using the information from the rest of the fields to train the algorithm.

### 3.3. Validation of RF Ability to Predict Whole-Plot Yield Using Data from the Remaining Fields

Studies such as those conducted by Pantazi et al. [40] or Kayad et al. [41] estimated the yield of specific pixels using different sources of information and algorithms. However, this study analyzed the potential of using satellite information and elevation (derived from a high-resolution DEM) to estimate the yield of an entire plot for which detailed yield information was not available. For this purpose, the RF algorithm was trained 13 times using the information of 12 parcels to estimate the yield of the remaining one (the "leave one out" validation technique). This process is not usual since few works have performed this process [41]. In this different validation, the highest mean $R^2$ was 0.60 while the lowest mean RMSE was 1223.32. These results were obtained using seven variables in each tree (MTRY = 7). Individual analyses of each plot showed that the $R^2$ ranged from 0.13 to 0.88 (Table 2) and the RMSE from 562.34 to 2229.17 (with these values corresponding to the Menor and Prado plots, respectively). However, the analysis showed that the predictive ability of RF to forecast an entire plot is weaker than that of predicting a single pixel. For example, the RMSE mean increases from 496 to 1200.49.

**Table 2.** Coefficient of determination ($R^2$) and root mean square error (RMSE) of the predicted yield of each entire plot (validation) using the information from the rest of the plots. MTRY is indicative of the number of variables used to create each tree.

| Plot | Coefficient of Determination ($R^2$) | | | Root mean Square Error (RMSE) | | |
|---|---|---|---|---|---|---|
| | MTRY = 2 | MTRY = 7 | MTRY = 14 | MTRY = 2 | MTRY = 7 | MTRY = 14 |
| Alto | 0.57 | 0.59 | 0.54 | 1694.20 | 1672.14 | 1646.13 |
| Apelarri | 0.27 | 0.42 | 0.46 | 1094.64 | 898.26 | 849.75 |
| Apelarri F | 0.86 | 0.88 | 0.88 | 1442.18 | 1332.08 | 1299.37 |
| Babea | 0.86 | 0.85 | 0.82 | 1142.54 | 1144.78 | 1104.33 |
| Barataua | 0.56 | 0.58 | 0.57 | 1070.95 | 1085.19 | 1101.12 |
| Iruleko | 0.57 | 0.60 | 0.62 | 1266.80 | 1285.71 | 1240.43 |
| Foronda | 0.70 | 0.68 | 0.65 | 919.62 | 1025.61 | 1129.08 |
| Kukura | 0.86 | 0.86 | 0.83 | 941.36 | 1085.36 | 1197.14 |
| Menor | 0.85 | 0.83 | 0.79 | 556.34 | 562.92 | 577.14 |
| Ollavarre | 0.21 | 0.13 | 0.01 | 1647.95 | 1473.98 | 1331.94 |
| Otatza | 0.60 | 0.59 | 0.56 | 872.88 | 850.42 | 962.20 |
| Parque | 0.38 | 0.39 | 0.39 | 968.63 | 960.87 | 999.55 |
| Prado | 0.32 | 0.47 | 0.49 | 1750.24 | 2229.17 | 2463.87 |

The cause of the worsening of the prediction capability could be the size of the dataset and plot heterogeneity. For example, Babea and Kukura are similar, so the algorithm trained with information from one of these has good capability to predict the other. The worst prediction was of the Ollavarre plot. Although the average production of this plot is similar to that of Parque, the two are probably not similar in terms of the auxiliary variables, and so the forecast was not good. However, when 70% of the data is used in the training process, some of the data will come from the Ollavarre plot. In the validation process, when the remaining 30% of the data information is used, there will be some similar data, allowing a better prediction. Despite this, the results obtained in the test validation of the whole plots are similar to those reported by Segarra et al. [42], where the RMSE% is around 20%. Considering these results, the RMSE% of the Torres plot prediction is expected to be close to 20%, although there is no certainty about this because RMSE variation is very high (Table 2). Therefore, it is essential to collaborate with the farmer and discuss with him the estimated yield map to see if it fits well with his yield knowledge acquired over years of fieldwork. The RF hyperparameters were adjusted to 500 trees and the MTRY to 7.

The calculation of the importance of the permutation characteristic of each prediction shows that the variable that contributes most to the model is the DEM. On the other hand, the images that contribute the most are those corresponding to the month of June (28 June and 18 June). Similar results were reported by Martí et al. [34]. In addition, the image corresponding to 19 February is the third largest contributor. This is not surprising, as

Hunt et al. [20] had already reported an increase in the correlation between NDVI and final yield during the tillering phase.

### 3.4. Fertilization Prescription Map

After validating the predictive ability of the RF algorithm to forecast yield for an entire plot and adjusting the hyperparameters to optimize its performance, it was run over the Torres plot (Figure 1, green dots) belonging to farmer 2. The mean estimated yield with the model was 5.3 t ha$^{-1}$, which is within the range of the production of the other plots (Table 1). The maximum and minimum values obtained from the model do not show anomalous data as they ranged from 7.4 ha$^{-1}$ to 4.3 t ha$^{-1}$. To facilitate the interpretation of the information by the end user, a continuous map without gaps was created by an interpolation using the ordinary kriging method (Figure 5A). The RMSE between the interpolated map and the measured values was 0.127. The interpolated map was considered good as there was no distortion compared to the original data. Figure 5A shows the estimated yield map once interpolated and classified into four different homogeneous zones using the ISODATA algorithm. The plot was divided into four different zones at the request of the farmer. Zone 1, with an average yield of 4.6 t ha$^{-1}$, was the least productive while zones 2 and 3 had average yields of 5.1 and 5.5 t ha$^{-1}$, respectively. With a production of more than 6 t ha$^{-1}$, zone 4 was the most productive (Figure 5A).

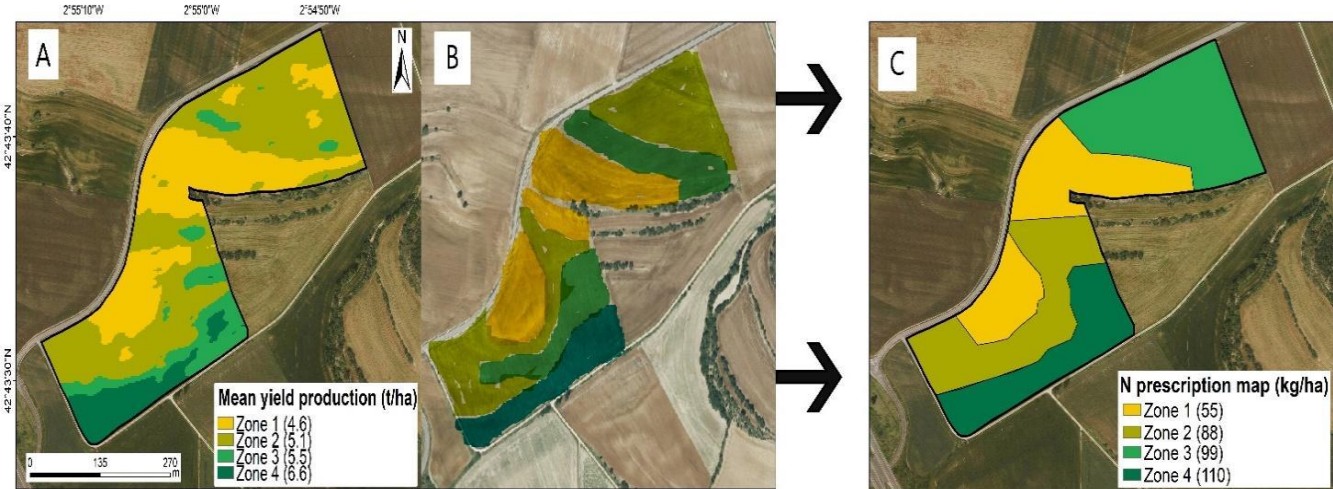

**Figure 5.** (**A**) Interpolated map based on the estimated yield data according to the RF algorithm. (**B**) Yield map provided by the farmer. (**C**) Nitrogen fertilizer prescription map derived from maps (**A,B**). Afterwards, to verify whether there were differences in production between the zones, an ANOVA, and subsequent post hoc HSD Tukey test were conducted. The analysis showed that there were significant differences ($p < 0.001$) in production between the four zones (**A**).

While the predictive ability for a whole plot was validated (Table 2), the results obtained showed that the prediction accuracy was variable. Detailed yield information is usually not available because few smallholder farmers have a yield monitor [43], and so the use of common metrics ($R^2$, RMSE, accuracy, Kappa index) to validate the result has to be discarded. To try to solve this handicap and in view of the farmer's in-depth knowledge of the field, he was invited to represent the yield map that he had in mind (Figure 5B) before showing him the map obtained using the model. Previous works published by Martínez-Casasnovas et al. [44] have shown the usefulness of this type of information. A simple visual comparison of both maps shows similar production patterns, strengthening the accuracy of the results obtained in the validation of the algorithm for the whole parcel in the previous section. Seeing the similarities between the map obtained by the algorithm and the one presented by farmer 2, he was asked about how he was able to differentiate the

production areas so well. His surprising answer was that that the noise of the grain hitting the hopper was much louder in the higher production areas.

These models work with high-resolution data that provide information on the current state of the crop. However, they do not supply information on the cause of yield differences. Collaborative work with the farmer allows scientists to understand the reasons behind yield variability. In this case, the southernmost part of zone 1 is characterized by shallow and pale-colored surface soils and as being the least productive area of the field (Figure 5A). Due to the smaller soil volume and pale surface color, it heats up faster, and, for this reason, cereals tend to sprout abundantly. However, because of a low soil water-holding capacity, the crop does not grow adequately later due to water stress. The farmer indicated that the origin of the low production of the northern part of zone 1 was different (Figure 5A). Until 30 years ago, this zone was a "ribazo" (an area characterized by a steep slope between two plots). This area was filled in with low-quality soil during the operation conducted to allow agricultural mechanization. This plot comes from the union of several smaller plots that have been combined over the years. This combination process is common in the area and can be verified by studying old orthophotos that are available from 1945 and by gathering information from farmers in the area. This line of work is interesting and would very probably provide valuable information and knowledge to be able to estimate the cereal yield in this and other plots.

Furthermore, the farmer confirmed that the high-yielding zone 4 (Figure 5A) was in a deeper and colder soil and is situated next to a stream, thus having a greater soil volume that holds more water. Usually, the crop has sprouting difficulties because of excess water but afterwards tends to grows better because the crop has more water available. This is a clear example of the knowledge held by the farmer about his field and his comprehension of the surrounding environment.

After comparing the map obtained with the machine learning algorithm (Figure 5A) and the map provided by the farmer (Figure 5B), an N fertilizer prescription map (Figure 5C) was agreed upon with the farmer. The zones of the prescription map must be large enough to work with a fertilizer spreader with a working width of 24 m. The fertilizer used was CAN in granular form, which has a nitrogen concentration of 27%. It was applied following the prescription map (Figure 5C) using an Amazone ZA fertilizer spreader with section cutting. Fertilizer doses were established based on the potential yield obtained by RF. Soil Nmin was not considered in this study for several reasons. Firstly, the spatial and temporal variability of Nmin in the soil is high [45], and the time and effort required to obtain an accurate sampling is completely unaffordable in commercial exploitations. For example, the Association of German Agricultural Research Institutes recommends taking 15 soil samples every 90 m$^2$ to capture Nmin variability. Considering the size of this plot, this would mean 1333 sampling points. Secondly, the soil Nmin value can change rapidly depending on leaching, mineralization, and gaseous losses. Thirdly, previous studies performed in the study area showed that Nmin values at the beginning of winter were low [46], always below 60 kg N ha$^{-1}$. Regarding crop uptake, ARVALIS, the French arable crops R&D institute (https://www.arvalis-infos.fr/les-besoins-unitaires-en-azote-des-varietes-reactualises-pour-2020-@/view-14925-arvarticle.html (22 September 2022)), measured that wheat extracts between 28 and 32 kg of N per ton of yield. With this data as a benchmark, a nitrogen recommendation was made in consultation with the farmer and based on a yield prescription map. The lowest fertilizer dose, 55 kg N ha$^{-1}$ as top-dressing, was assigned to the least productive area (zone 1) while the usual top-dressing dose of 110 kg N ha$^{-1}$ was maintained for the most productive area (zone 4), and intermediate values (88 and 99 kg ha$^{-1}$) for the other two zones (Table 3). Specifically, the amount of N applied in the least productive areas was half that of the most productive.

**Table 3.** Dose of N applied in the plot with and without zoning.

| Variable N Application | | | | Homogeneous N Application | | | |
|---|---|---|---|---|---|---|---|
| Zone | kg N ha$^{-1}$ | Area (ha) | kg N | Zone | kg N ha$^{-1}$ | Area (ha) | kg N |
| 1 | 55 | 3.37 | 185 | 1,2,3,4 | 110 | 12.23 | 1345 |
| 2 | 88 | 2.95 | 259 | | | | |
| 3 | 99 | 3.70 | 375 | | | | |
| 4 | 110 | 2.21 | 244 | | | | |
| | | 12.23 | 1064 | | | 12.23 | 1345 |

*3.5. Variable Fertilizer Application Considerations and On-Farm Experimentation*

To fertilize the plot homogeneously with a conventional dose of 110 kg N ha$^{-1}$ would require 1345 kg (Table 3, final column). However, zone-based fertilization allowed a reduction in the N supply to 1064 kg (Table 3, second column). This equates to a reduction of 281 kg N, about 20.8%, thus achieving the new goal of reducing fertilizer consumption by 20% as established by the EU in its From Farm to Fork strategy. Moreover, in 2020, when this experiment was carried out, the price of ANC 27% fertilizer was 0.22 € kg N$^{-1}$ [47]. The use of the VRF technique allowed a saving of 231.3 € of fertilizer in this plot. However, the price of fertilizers is increasing dramatically and the use of this technique, in addition to contributing to reducing the impact of agriculture on the environment, would allow the farmer to increase the economic return of the farm. The reduction of the environmental impact comes from a better utilization of the nutrients by crops. The application of a higher dose in the more productive areas and a lower dose in the less productive ones [48] is expected to reduce fertilizer losses [49].

Crop vigor monitoring with the S2 NDVI showed that, as the crop cycle advances, the variability of crop vigor increasingly resembled the fertilization prescription map (Figure 5C). This trend was maintained until the onset of plant senescence (GS80). Since a yield monitor was not available in this plot, the farmer was asked if he had seen any change in the yield. The farmer's response was negative, confirming that the VRF had not decreased the expected production. Even the farmer commented that, in less productive zones, the yield might have been somewhat better. These results are similar to those obtained by Argento et al. [50] when using variable fertilization. There have been glimpses of the economic and environmental improvements that the use of VRF techniques could bring since the beginning of precision agriculture [51]. However, the latest study published by Späti et al. [52] shows that the net economic return obtained by small farmers adopting this technique is related to soil variability, fertilizer price, and the cost of the sensor used to make measurements. The cost of fertilizer fluctuates from season to season depending on the price of energy and fuel [53]. High fertilizer prices reduce the farmer's profit margin, which means an increase in the effort the farmer has to make [54]. Rising energy prices have caused the price of fertilizer to triple from the 2021 season to the 2022 season [47]. However, it has opened the door to the adoption of variable rate application technology because of its ability to optimize fertilizer use without decreasing yield. Bearing in mind socio-economic circumstances and new technological development capabilities, it is essential to reinforce collaborative work between academics and farmers with a view to integrating agronomic knowledge into new technological developments.

However, one of the main barriers to a more fluid collaboration between farmers and scientists is the experimental design used by researchers and the tendency of scientists to generalize the results obtained in experimental plots. Close collaboration with farmers would make it possible to evaluate the spatiotemporal variability of their plots in a less costly and more efficient way than if this had to be achieved employing classical controlled field trials. As reported by Laurent et al. [55], in this type of trial, the differences caused by spatiotemporal variability may be greater than those caused by the different treatments. To avoid this tendency and to obtain more realistic and scalable results, the trial should be conducted using commercial plots. OFE considers the heterogeneity of farmers' circum-

stances, practices, and needs, providing practical and contextualized information on how to use, adapt, and develop local innovations [56]. The adaptation of the technique used to the specific needs of the plot was a work carried out jointly with the farmer.

Once the experimentation was concluded, the results obtained through this technique were visible to the farmer. After the knowledge is transferred, replication of the experiment is simple since the farmer now has both the resources and the knowledge [57]. For example, the close collaborative work between academics and the farmer carried out in this work encouraged the farmer to extend the use of the VRF to his entire farm, which consists of more than 70 plots and 200 ha. According to the farmer, the need-to-know GIS tools to edit fertilizer prescription maps constitute the bottleneck of the process. According to the farmer, if this step, which allows easy map editing, is not solved, widespread implementation of this technique would be extremely low or non-existent in our conditions. To try to solve this problem, the fertilizer prescription maps for the following season (2020–2021) were made using the remote sensing decision support tool implemented in the Agrogestor platform (www.agrogestor.es (22 September 2022)). The procedure to obtain a prescription map is as follows. Firstly, the farmer sets the plot and dates, then the platform provides maps of vegetative indices such as the NDVI. In the next step, the farmer must select the map that most closely resembles yield variability, and then the platform classifies the map into four zones using an unsupervised classification algorithm. Using this map, the Agrogestor platform allows users to easily create a fertilizer prescription map. The platform provides a classified map where the farmer allocates a fertilizer dose to each zone. The platform can export the prescription map to a spreader-friendly format. The farmer confirmed that the process of using the tool was easy and can help in the adoption of this technique.

Another advantage of performing trials with the farmers is the added knowledge generated on the social level, through knowledge sharing and updating between both groups [58]. In addition, at the scientific level, analytical work through meta-analysis and integration of real data helps to better address real problems. The integration of innovative technologies in the agricultural sector could help solve the problem of an aging sector as many young people are not attracted to agriculture, considering it old-fashioned and obsolete [59]. The technification of the agricultural sector through the acquisition of modern technologies may reverse this process by turning what is considered obsolete into something new.

The recent development of online farming games can also help make agriculture more attractive, showing farming through technology that young people are comfortable with. The adoption of such technologies can help to reduce the technical, legislative, economic, and educational barriers that currently exist [60]. An economically, socially, and environmentally sustainable agri-food sector can help to repopulate rural areas, making them more attractive, lively, and dynamic, and paying special attention to the generation of wealth and the integration of young people and women [61].

Finally, a special mention should be made of the use of an innovative approach and close collaboration between farmers and academics to solve a critical problem of smallholder farmers who are being forced to reduce the consumption of fertilizers by both European legislation and rising prices. The methodology followed in this study is therefore of interest because it allows the farmer to better adjust fertilization to the needs of the plant, especially in situations where the potential yield is low due to lack of water. This is something that the farmer participating in the study continuously highlighted. Although this work has been carried out thanks to the help of only two farmers, it should be noted that they are both benchmarks in the sector, pioneer farmers who can act as tractors for other farmers who are interested in fertilizing more rationally. The flow of information was bi-directional and enriching for both groups. The farmer was provided with the knowledge necessary to implement this technique while the farmer provided local knowledge of the variables that directly affect the crop. Many of these variables can be measured and their integration into future algorithms has been considered for a future project. In addition, the use of machine learning algorithms based on information from other farmers and free

auxiliary information (S2 and DEM) to solve the problem of the lack of yield information for a plot was also novel.

## 4. Conclusions

Combining machine learning techniques and a farmer's knowledge of his plot, it was possible to establish a fertilization map of a plot that allowed significant savings in fertilizer consumption.

Of the 15 machine learning algorithms that were tested, RF performed best. This study corroborated the usefulness of free auxiliary information such as S2-derived vegetative indices and LiDAR-derived elevation. In addition, the predictive ability of this algorithm to predict the yield of an entire plot using the "leave one out" technique was tested. The collaboration between scientists and farmer resulted in a very positive exchange of information that allowed the farmer to change the fertilization strategy of the whole farm and the scientists to better understand how soil properties and plot history affect yield. All this information could be used to improve future models.

**Author Contributions:** A.U. worked in the following: Conceptualization, Methodology, Software, Data processing, Formal analysis original draft preparation, Visualization, Investigation, Interpretation. A.C. worked in the following: Methodology, Data acquisition, Results analysis. Resources. I.E. worked in the following: Data acquisition. A.A. worked in the following: Methodology, Writing—Reviewing and Editing, Supervision of Parameter computing, Funding acquisition, Project administration. All authors have read and agreed to the published version of the manuscript.

**Funding:** This work was funded by the BIKAINTEK grant of the Basque Government, Department of Economic Development, Sustainability and Environment. It was also supported by the NITRALDA Project founded by the Basque Government in the "Strategy for the digitalization of the food value chain and gastronomy in the Basque Country". It was additionally supported by the LIFE program thanks to the project AGROGESTOR (LIFE16 ENV/ES/287).

**Data Availability Statement:** Data available in a publicly accessible repository that does not issue DOIs. The Topographic Data can be found in https://www.geo.euskadi.eus/cartografia/DatosDescarga/, accessed on 23 August 2022. The satellite information data can be found in https://scihub.copernicus.eu/dhus/#/home accessed on 23 August 2022.

**Acknowledgments:** The authors would like to thank Hermanos Torre, from the Unión de Agricultores y Ganaderos de Álava (UAGA) and Javier Alava, a farmer in the GARLAN cooperative, for providing the possibility to carry out the research in their plots and giving us yield information. The HAZI Fundazioa is thanked for its support in the project. Gerardo Besga is thanked for his work in reading the latest version and the improvements proposed to make the text more understandable.

**Conflicts of Interest:** The authors declare no conflict of interest. The funders had no role in the design of the study, in the collection, analyses, or interpretation of data, in the writing of the manuscript, or in the decision to publish the results.

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
