# Peer review of "Intra-Plot Variable N Fertilization in Winter Wheat through Machine Learning and Farmer Knowledge"

_agronomy, doi:10.3390/agronomy12102276_

Round 1

Reviewer 1 Report

Overall, I believe the work here is scientifically sound and of interest to readers.

I feel the paper is missing some important details in the methods section.  

List specific red and NIR wavelengths and resolution for Sentinel2 or reference others.

It seems the authors state that they have 14 sequential sets of imagery, but then 4 specific wheat stages are mentioned.  It's unclear exactly what imagery was included in the yield estimation model or how it was combined/averaged (or whatever happened).  There are also 14 fields which further complicates this. It would help for the authors to be completely descriptive and use language to separate fields and images. 

If spectral data were combined, why?  I would have suspected that the DEM data combined with one or two imagery dates would have provided the best estimate, especially the variation in the early and late readings. 

The authors discuss lowest and highest RMSE values, but without much context.  It's hard to know what is (and what the authors consider) biologically significant. 

The authors make quite a bit about how collaboration with farmers is valuable and i agree.  But for what i'm reading, there is only one farmer and really one field where this concept was applied.  If that is true, I think the authors are overstating what was actually accomplished here.  It's not an extensive study with many farmers so may be a stretch to say this approach can 'solve a critical problem of smallholder farmers'.  

Author Response

First of all, we would like to thank the reviewer 1 for his effort to read and improve the document. The comments made by the reviewer are greatly valued and we have tried to address all of them below. Changes have been done in the reviewed manuscript according to these comments and observations, which for sure have served to improve the paper.

Reviewer 2 Report

Dear authors,

Thank you for the opportunity to review this Manuscript (Intra-plot variable N fertilization in winter wheat through machine learning and farmer knowledge).  The study has great results and demonstrates that the combining machine learning techniques and the farmer's knowledge of his plot, it was possible to establish a fertilization map of a plot that allowed significant savings in fertilizer consumption. There is some aspect that should be reviewed by authors, but the Manuscript is well-written.

INTRODUCTION

The first paragraph was very interesting, and demonstrate the importance of farmer, agronomic advisors, together with scientists. The authors could add results of studies with information of this interaction.

Line 60, describe the variability of the plot.

Line 59, explain the N losses with different N sources.

Line 61, give more details about the “Accurate and early yield estimation is a good way to adapt fertilization to plant requirements.”

Line 63, “the beginning of stem elongation” For each crop?

Which is “twin S2 A+B satellites”?

The objective is confused, check it.

MATERIAL AND METHODS

Describe the size of each plot

Explain the historic of each area

Add information of plant management.

Is there soil pH correction? Soil characterization?

Soil texture?

RESULTS AND DISCUSSION

In Figure 3, explain the treatment “r”

The information in the Figure 3, is not clear. Explain the idea of the figure in the text. Maybe, the Figure 3 could be edited.

Line 263: I did not understand the code “GS31”, for example. Check it

Explain, the difference between LM and LM MDT for RMSE and Rsquared. The difference was intense.
Explain, the superior performance of
RF

Lines 407-409: “Until 30 years ago, this zone was a "ribazo" (an area characterized by a steep slope between two plots). This area was filled in with low-quality soil during  the operation conducted to allow agricultural mechanization.” The historic of area is very important to understand the dynamic of N in soil.

Author Response

First of all, we would like to thank the reviewer 2for for his effort to read and improve the document. The comments made by the reviewer are greatly valued and we have tried to address all of them below. Changes have been done in the reviewed manuscript according to these comments and observations, which for sure have served to improve the paper.
